# Partition and conquer: Multimodal Autoregressive learning for time-aligned and contextual modalities

## Abstract

One of the main challenges of multimodal learning is the need to combine heterogeneous modalities (e.g., video, audio, text). For example, video and audio are obtained at much higher rates than text and are roughly aligned in time. They are often not synchronized with text, which comes as a global context, e.g. a description. Furthermore, video and audio inputs are of much larger volumes, and grow as the video length increases, which naturally requires more compute dedicated to these modalities and makes modeling of long-range dependencies harder.

We here decouple the multimodal modeling, dividing it into separate, focused autoregressive models, processing the inputs according to the characteristics of the modalities. We propose a multimodal model consisting of an autoregressive component for the time-synchronized modalities (audio and video), and an autoregressive component for modalities which are not necessarily aligned in time but are still sequential. To address the long-sequences of the video-audio inputs, we propose to further partition the video and audio sequences in consecutive snippets and autoregressively process their representations. To that end, we propose a Combiner mechanism, which models the audio-video information jointly within a timeframe. The Combiner learns to extract audio and video features from raw spatio-temporal signals, and then learns to fuse these features producing compact but expressive representations per snippet. Our approach achieves the state-of-the-art on well established multimodal benchmarks, outperforming much larger models. It effectively addresses the high computational demand of media inputs by both learning compact representations, controlling the sequence length of the audio-video feature representations, and modeling their dependencies in time.

## 1 Introduction

Multimodal models aim to combine the signals from multiple varied sources, which makes them both universal and useful for practical applications. However, these modalities have diverse characteristics and are challenging to combine uniformly by a single model. For example, video and text have disparate sampling rates: a video has many frames per second, but text or other similar types of global context, e.g., a description or title, can be provided once per video, or asynchronously to the video. Video also takes a larger portion of the input. At the same time, video and audio are naturally co-occurring and appear (almost) synchronously. They are roughly aligned and complementary. This co-occurrence in time can contribute to their joint learning and serve as a rich self-supervisory learning signal, applied more frequently than global text signals. So, ideally, these modalities need to be processed by differently-synchronized model components, which process more adequately inputs of different frequencies and allocate more parameters to the more abundant modalities.

Following the success of large language models, where text input sequences are processed autoregressively, many recent multimodal models reuse the autoregressive text models, feeding in other modalities, e.g., as embeddings, (Chen et al., 2023a; Wang et al., 2022b; Piergiovanni et al., 2022a; Li et al., 2021a; 2022b; Alayrac et al., 2022), or by tokenizing the visual inputs to be processed together with the text token sequence (Wang et al., 2022d; Aghajanyan et al., 2022; Yu et al., 2023; Reed et al., 2022; Zellers et al., 2022)). However, the imbalance of the information volume is large and models which are well suited to encode/decode text sequences process only highly compressed

image or video features (Alayrac et al., 2022; Nagrani et al., 2021). For example, the Flamingo model (Alayrac et al., 2022), subsamples the video features significantly, dedicating only about 1% of the parameters to the image and video inputs, leaving the rest for text processing. Alternatively, methods that process the video running each frame independently through an encoder or tokenizer, reach an upper bound on the number of frames quickly (Yan et al., 2022; Piergiovanni et al., 2023b). For longer inputs, these representations are insufficient to properly represent the modalities, which inherently limits the ability to model fine-grained or long-range dependencies.

We here propose an audio-video-text multimodal model where we decouple the autoregressive modeling into a component for time-aligned modalities, e.g., audio and video, which are processed in time, autoregressively, and an autoregressive component for non-time-aligned contextual modalities e.g., text. Cross-attention weights coordinate the learning between these components. This decoupling allows for better parameter distribution within the model, allocating sufficient capacity for the media modalities, and leads to smaller models overall. Furthermore, we propose to partition the time-aligned modalities into time segments, where audio-video joint representations are learned and accumulated in time. To that end, we introduce a joint feature learning mechanism for the media modalities, called the Combiner, which fuses their features and produces a more compact representation. We extract low level spatio-temporal representation from the raw media inputs in order to capture the dynamic nature of videos in a high-level representation and combine it with audio features within concurrent timesteps. Our model enables consuming multimodal inputs at different rates and allows for larger inputs for longer videos. The Combiner effectively balances the need for efficient media representations and ones which are expressive enough to preserve the media content. It sufficiently represents the events and activities in the videos and other concurrent modalities and can be handled by subsequent autoregressive models, which allows for learning long-range dependencies. Our contributions are:

- An autoregressive multimodal model, subdividing learning into autoregressive modeling for time-aligned media modalities and non-time-aligned contextual modalities.
- Joint feature representation learning via the Combiner to balance learning of efficient media representations which are also sufficiently expressive to preserve the media content.
- We demonstrate learning with 128-512 video frames without increase in model parameters. This is in contrast to prior works that use 32 (Gao et al., 2023) or 8 frames (Yan et al., 2022).

Our model outperforms the state-of-the-art on multiple benchmarks, with large margins on audio-video-text datasets and on long video datasets. It outperforms much larger models, as well.

## 2 RELATED WORK

Architectures for video-language understanding commonly use a joint transformer, where video inputs are fed in together with text tokens and processed autoregressively (Fu et al., 2021; Zellers et al., 2021)). This is often accomplished with tokenizing the visual inputs. Video-text pretraining approaches (Miech et al., 2019; 2020; Wang et al., 2022e;c; Li et al., 2023; 2022a) use masked token modeling and reconstruction (Fu et al., 2021), masking with cross-attention on multimodal inputs (Alayrac et al., 2022), or contrastive learning (Wang et al., 2022c; Xu et al., 2023; Zellers et al., 2021; Dong et al., 2023). Visual synthesis models have extensively used autoregressive models, by learning to generate pixel-wise predictions (van den Oord et al., 2016), or by learned discrete tokens from images or videos, e.g., NÜWA (Wu et al., 2022), VideoGPT (Yan et al., 2021), GODIVA (Wu et al., 2021). In other models, encoder-decoder or decoder-only architectures extend an image-text model to a video-text one (Wang et al., 2022b; Yan et al., 2022), where video is processed by individual frames which are then combined. Some architectures instead extract full video signals (typically as embeddings) before feeding them to the model (Xu et al., 2023). Another option is to attach a projection or re-tokenization layers e.g., as in Perceiver in Flamingo (Alayrac et al., 2022), to reduce the amount of visual tokens added to the model. Our approach differs substantially, as the media input features have a specifically designed component to learn them jointly and in time, producing more abstract representations, suitable for modeling long videos.

Multimodal audio-video-text models have also gained popularity: UAVM (Gong et al., 2022) propose joint learning of audio and video by building invariant transformer module which can be reused by either signal. Multimodal Transformer (Tsai et al., 2021) proposes cross-attention mechanisms,

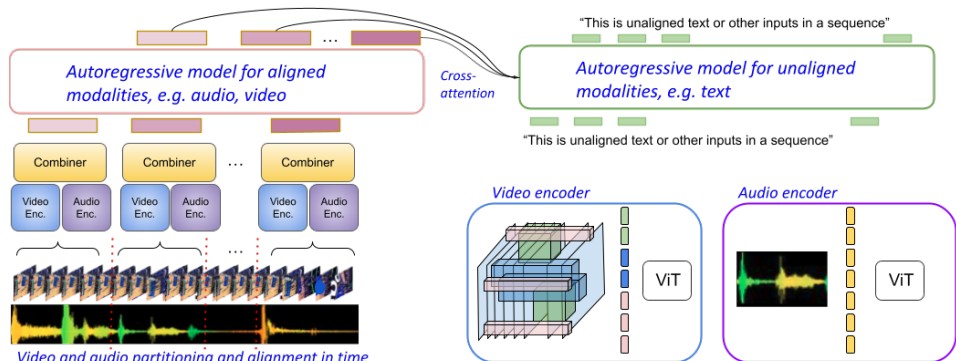

Figure 1: The model architecture consists of an autoregressive model for the time-aligned modalities, such as audio and video, which are partitioned in chunks (left) and an autoregressive model for the unaligned context modalities, which are still sequential, e.g., text (right). This allows adequate computational capacity to the video/audio time-synchronized inputs, including processing them in time autoregressively, before fusing with the autoregressive decoder for unaligned text (right). Joint feature learning is conducted by the Combiner, balancing the need for compact representations and allowing sufficiently informative features to be processed in time.

for cross-modal learning on all pairs of video-audio-text data, which Pellegrain et al. (2021) extends to longer sequences. Zellers et al. (2022) demonstrate joint multimodal audio-video-text learning but only aligning text and audio. Gong et al. (2023) use contrastive audio-video learning, whereas Huang et al. use masked autoencoder for audio-video learning. Both approaches tokenize the audio video inputs independently in 2D patches which are used for further processing. Rouditchenko et al. (2021) and Korbar et al. (2018), propose contrastive learning for audio-video signals, leveraging the time-alignment between them. Audio-video late fusion is also common, (Pibre et al., 2022).

Our work is related to long-form video understanding (Wu & Krahenbuhl, 2021; Sun et al., 2022). Long-form videos have been handled by hierarchical feature learning e.g., the Temporal Window Attention (Sun et al., 2022) where dependencies are learned locally and then further propagated to higher level cross-attention modules. Ashutosh et al. (2023) propose contrastive learning at different hierarchical levels. Gao et al. (2023) segment videos then pool their features into a small representation. Memory-augmented model for long videos (Wu et al., 2023) accumulate prior context in learnable 'memory', to be referenced at each step of learning. Our work contributes by proposing a balanced approach of locally learning important features, jointly within the modalities.

## 3 APPROACH

Autoregressive models are powerful generative models that are well suited for data which appears in a sequence, modeling the probability of the current value, conditioned of previous ones. Video and audio information is sequential but also roughly time-synchronized. At the same time, other modalities e.g., text, might be provided globally per video as context and applied to the full video rather than to specific parts[1]. To address the challenges of modeling diverse multimodal inputs, we propose to subdivide the autoregressive modeling by learning separate autoregressive models: one for the time-aligned modalities (audio-video), Section 3.3, and another one for modalities which are not necessarily aligned in time but are still sequential, Section 3.4. Learning across these is coordinated by cross-attention mechanisms, where here the media inputs (audio-video) are allocated a lot more parameters and are properly modeled in time. A learning module, called the Combiner (Section 3.2), combines the lower-level signals from video/audio snippets. Here information is processed spatio-temporally, extracting features particularly relevant to dynamic changes in the inputs.

**Architecture overview.** At a high level, the architecture consists of two main learning components (Figure 1): The first one is an autoregressive component which is designed to process (almost) synchronized multimedia inputs e.g., video+audio and combine their inputs in time (Figure 2, left).

---

[1]Text, e.g., ASR, might also appear concurrently with audio/video and can contribute to improved understanding of the video content. We leave this to future work.

In order to process the video and audio signals, and to accommodate longer video/audio inputs, they are partitioned into smaller chunks (roughly synchronized in time) for which a joint audio-visual representation is learned via the Combiner as described below. The second component processes the context, or the signals not aligned in time, such as global text information, which are often still sequential. It is autoregressive as well, and uses the combined latent space as cross-attention inputs.

**Problem formulation:** We have an input video sequence of $N$ frames $\boldsymbol{v} = \{\boldsymbol{v}_1^f, \boldsymbol{v}_2^f, \ldots \boldsymbol{v}_N^f\}$, and audio wave signal of $M$ timesteps $\boldsymbol{a} = \{\boldsymbol{a}_1^f, \boldsymbol{a}_2^f, \ldots \boldsymbol{a}_M^f\}$, where the audio signal is captured during the duration of the video and corresponds to the given video input. Additionally we have an input text sequence $\boldsymbol{t} = \{\boldsymbol{t}_1^f, \boldsymbol{t}_2^f, \ldots \boldsymbol{t}_P^f\}$, which is related to the video/audio and might vary according to the tasks e.g., it can be a description, a question-answer pair, meta information and etc.

**Partitioning of the media inputs:** In order to process the video sequence efficiently and to learn the correlation of features in time, we partition the input video into into $T$ non-overlapping segments or chunks, with $\boldsymbol{v}_t$ and $\boldsymbol{a}_t$ denoting the video and audio input per segment. Here each chunk captures all input data between two timestamps (i.e., video and audio snippets), as follows:

$$\underbrace{\boldsymbol{v}_1^f, \boldsymbol{v}_2^f, \ldots, \boldsymbol{v}_{N/T}^f}_{\boldsymbol{v}_1}, \underbrace{\boldsymbol{v}_{N/T+1}^f, \boldsymbol{v}_{N/T+2}^f, \ldots, \boldsymbol{v}_{2N/T}^f}_{\boldsymbol{v}_2}, \ldots \underbrace{\boldsymbol{v}_{(T-1)N/T+1}^f, \boldsymbol{v}_{(T-1)N/T+2}^f, \ldots, \boldsymbol{v}_N^f}_{\boldsymbol{v}_T}, \quad (1)$$

Thus the video is represented by its chunks instead, $\boldsymbol{v} = \{\boldsymbol{v}_1, \boldsymbol{v}_2, \ldots \boldsymbol{v}_T\}$, and more specifically latent features will be extracted from each chunk to represent the video (as described in Section 3.1). A similar partitioning is done for the audio signals, where they are partitioned in $T$ chunks to be synchronized in time to the video chunks, $\boldsymbol{a} = \{\boldsymbol{a}_1, \boldsymbol{a}_2, \ldots \boldsymbol{a}_T\}$. Here too we assume that audio features will be extracted from the raw audio signal, as described in Section 3.1.

### 3.1 Audio/video features

**Video features.** Prior models captured video information frame-by-frame which lacks the temporal information, essential to video understanding and might miss dynamic events. Alternatively, 3D convolutions (Wu et al., 2022), sparse 3D tubes (Piergiovanni et al., 2023a) and others learn spatio-temporally which can capture key dynamic changes in the videos. We expand on these ideas and extract sparse 3D tubes (Piergiovanni et al., 2023a) from the videos which span all 3 dimensions of the video snippet starting at various locations. The sparse 3D tubes, together with standard 2D patches are processed via a ViT encoder. Rather than applying offsets, the tubes are applied at the beginning of the snippet. Per video chunk $t$ we denote $\hat{\boldsymbol{v}}_t$ as the time-aligned features for this chunk, and thus $\hat{\boldsymbol{v}} = \{\hat{\boldsymbol{v}}_1, \hat{\boldsymbol{v}}_2, \ldots, \hat{\boldsymbol{v}}_T)$ are the time-aligned video representations for the whole video.

**Audio features.** Audio inputs arrive at a predefined frequency and can be processed in various ways. We here represent the audio as a spectrogram. The spectrogram is created so that the time bands match the 25 frames per second used in the videos, and thus can easily be split into snippets aligned with the video. The spectrogram for each snippet is processed by a ViT model, after an audio input projection layer. The ViT backbone is the same as the one used for video features. Reusing the visual component is previously shown to be advantageous (Gong et al., 2022). Similarly to above, we denote $\hat{\boldsymbol{a}}_t$ to be the audio feature per chunk $t$ and $\hat{\boldsymbol{a}} = \{\hat{\boldsymbol{a}}_1, \hat{\boldsymbol{a}}_2, \ldots, \hat{\boldsymbol{a}}_T)$ for the full video.

### 3.2 Modality Combiner

The task of the Combiner module is two-fold: 1) to combine the video (and audio) features at a specific snippet of time, learning their joint representation and 2) effectively compress the representation from each video/audio snippet, which allows our model to scale to longer videos.

When partitioning the inputs, the features for each modality, video and audio in this case, are (roughly) time-aligned latent features $\hat{\boldsymbol{v}} = \{\hat{\boldsymbol{v}}_1, \hat{\boldsymbol{v}}_2, \ldots, \hat{\boldsymbol{v}}_T)$ and $\hat{\boldsymbol{a}} = \{\hat{\boldsymbol{a}}_1, \hat{\boldsymbol{a}}_2, \ldots, \hat{\boldsymbol{a}}_T)$, where the maximum timestamp for any data incorporated into $\hat{\boldsymbol{v}}_t$ or $\hat{\boldsymbol{a}}_t$ is less than the minimum timestamp of any data incorporated into $\hat{\boldsymbol{v}}_{t+1}$ or $\hat{\boldsymbol{a}}_{t+1}$. Explicitly $\hat{\boldsymbol{v}}_t$ is composed of $f$ features of size $d$ giving it a shape of $(f, d)$ and $\hat{\boldsymbol{a}}_t$ is composed of $s$ features also of size $d$ with shape $(s, d)$. The role of the combiner is to map such time-aligned modal latent features into a smaller set of shared latent features. Specifically let $\hat{\boldsymbol{c}} = \{\hat{\boldsymbol{c}}_1, \hat{\boldsymbol{c}}_2, \ldots, \hat{\boldsymbol{c}}_T\}$ where $\hat{\boldsymbol{c}}_t = (\hat{\boldsymbol{v}}_t, \hat{\boldsymbol{a}}_t)$ having size $(n, d)$ and $n = f + s$ be the set of all time-aligned features from all modalities. The combiner then maps $\hat{\boldsymbol{c}}$ to a shared latent feature space $\boldsymbol{x} = \{\boldsymbol{x}_1, \boldsymbol{x}_2, \ldots, \boldsymbol{x}_T\}$ where $\boldsymbol{x}_t$ has shape $(m, d)$ where $n >> m$.

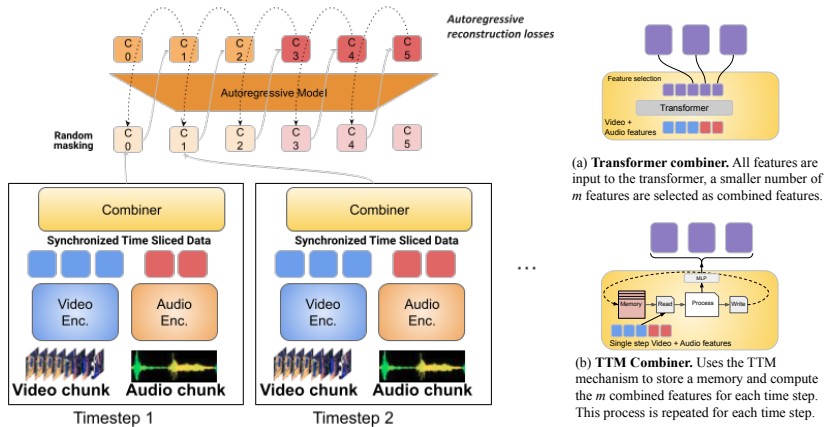

Figure 2: Autoregressive modeling of video and audio in time (left). Combiners (right).

Since the features produced by the combiner are going to be used in the sequential autoregressive modeling of video/audio, it is important for the combiner to not break causality thus:

$$\boldsymbol{x}_t = \text{Combiner}(\hat{\boldsymbol{c}}_1, \hat{\boldsymbol{c}}_2, \ldots, \hat{\boldsymbol{c}}_t) \tag{2}$$

We utilize two different architectures for the combiner, a standard Transformer one and a memory based approach, based on the Token Turing Machines (Ryoo et al., 2023), to reduce memory.

**Causal Transformer Combiner.** We explore a straightforward version of the Combiner, which consists of a standard Transformer model, here of $K$ layers (here, 8). For each step $t$ it maps the original set of features $\hat{\boldsymbol{c}}_t$ to $\boldsymbol{x}_t$ where $\boldsymbol{x}_t$ is of much lower dimensionality, i.e., effectively reducing the number of tokens (here $m = 32$) (Figure 2, right). The inputs to the Combiner are the latent features of the video and audio, which are concatenated before being fed to the Combiner. We here specifically implement a causal version of the transformer as it masks out inputs from future timestamps (i.e., $t$). The attention mechanism of the transformer is modified to mask features at the time-chunk level as described later in Section 3.3.1 (using Equation 8), thus all features from $\hat{\boldsymbol{c}}_t$ and the previous time steps are used to compute each output feature in $\boldsymbol{x}_t$ as in Equation 2. This effectively applies attention mechanisms to all the modality inputs jointly while respecting causality.

**Token Turing Machine Combiner.** Token Turing Machine (TTM) (Ryoo et al., 2023) is a recurrent sequential model with Transformers and token-based operations. It maintains an external 'memory' $M_t$ as a set of features, and updates it at every time step by reading and writing. Given inputs $\hat{\boldsymbol{c}}_t$ at each time step, it first 'reads' features to be processed, from input features as well as memory features. Such features, $\boldsymbol{z}_t$, are passed to the 'processor', which is implemented as a standard Transformer, generating a set of intermediate output features $\boldsymbol{o}_t$. These intermediate outputs are then used to update $M_t$ (i.e., memory 'write') as well as to produce the final output $\boldsymbol{x}_t$.

$$\boldsymbol{z}_t = \text{Read}(\hat{\boldsymbol{c}}_t, M_t) \tag{3}$$
$$\boldsymbol{o}_t = \text{Process}(\boldsymbol{z}_t) \tag{4}$$
$$M_{t+1} = \text{Write}(M_t, \boldsymbol{o}_t, \hat{\boldsymbol{c}}_t) \tag{5}$$
$$\boldsymbol{x}_t = \text{Output}(\boldsymbol{o}_t) \tag{6}$$

The key idea is to make the Transformer processor generate the outputs by utilizing memory $M_t$ instead of the entire history of features $\{\hat{\boldsymbol{c}}_1, \ldots, \hat{\boldsymbol{c}}_{t-1}\}$. Once trained, the differentiable read and write operations will be optimized to maintain $M_t$ so that it stores important features from the previous time steps $\{\hat{\boldsymbol{c}}_1, \ldots, \hat{\boldsymbol{c}}_{t-1}\}$ and updates it, at every step.

We implement TTM as the Combiner module to sequentially combine $\hat{\boldsymbol{c}}$. The function 'Process' is implemented with a standard Transformer with layers of multi-head self-attention and MLPs. The functions 'Read', 'Write', and 'Output' are implemented with TokenLearner (Ryoo et al., 2021) (which is similar to Perceiver (Jaegle et al., 2021) and attention pooling (Lee et al., 2019)). Note

that we are able to separately control the number of features in the memory as well as the number of 'Output' function features, allowing efficient combiner computation and feature generation.

The key advantage of the TTM Combiner is its utilization of memory features to sequentially process $\hat{c}_t$. The number of such memory features are much smaller than the total number of history features ($\{\hat{c}_1, \ldots, \hat{c}_{t-1}\}$) in general (e.g., 256 vs. $\sim$10k). This not only makes TTM a natural fit for the model, but also reduces the total time complexity of the TTM Combiner to be constant with respect to $t$, instead of $O(t)$ or $O(t^2)$ in Transformers. We observe that the TTM Combiner saves memory in both training and inference, about 30% less memory and reduces the runtime by about 18%.

### 3.3 Time-Aligned Video/Audio Autoregressive Modeling

We describe the autoregressive modeling of time-aligned video and audio. We apply autoregressive modeling strategy where we condition video/audio representations corresponding to a time interval on feature representations from previous time intervals. These representations are learned jointly by the Combiner, as described in Section 3.2. As mentioned, the video is first partitioned in $T$ smaller video snippets. Each of the snippets itself can be of size 4-64 frames (overlap is possible but currently not used). We extract spatio-temporal information into latent video features $\hat{v}_t$ and audio features $\hat{a}_t$ from the same video partition, apply Combiner to produce $x_t$. The feature representations per video chunk $x_t$ are then fed sequentially to the autoregressive model, where at each step we reconstruct the features from the previous step, conditioned on the prior inputs and the latent vector $h$ which corresponds to the latent representations learned within the autoregressive model:

$$p(\boldsymbol{v}, \boldsymbol{a}) = \prod_{t=1}^{T} p(\boldsymbol{v}_{t+1}, \boldsymbol{a}_{t+1}|\boldsymbol{h}_t)p(\boldsymbol{h}_t|\boldsymbol{x}_t)p(\boldsymbol{x}_t|\boldsymbol{v}_t, \boldsymbol{a}_t) \tag{7}$$

where $\{\boldsymbol{v}_1, \boldsymbol{v}_2, \ldots \boldsymbol{v}_T\}$, and $\{\boldsymbol{a}_1, \boldsymbol{a}_2, \ldots \boldsymbol{a}_T\}$ are the feature representations from the video and audio, $p(\boldsymbol{x}_{t-1}|\boldsymbol{v}_{t-1}, \boldsymbol{a}_{t-1})$ is estimated by the combiner, and $p(\boldsymbol{h}_{t-1}|\boldsymbol{x}_{t-1})$ is estimated by the latent causal model. This allows for learning from previous representations in the sequence (in time) and aims to predict the next-step feature representation (Figure 2, left). While autoregressive modeling has been used for videos and images, it is often done on a pixel-by-pixel basis (Weissenborn et al., 2020) which is highly inefficient and captures only short-term dependencies. With our approach, with autoregressive modeling and the Combiner, and we address both shortcomings.

**Latent Causal Modeling.** The autoregressive latent model estimates: $\prod_{t=1}^{T} p(\boldsymbol{h}_t|\boldsymbol{x}_t)$. This is done by applying an autoregressive transformer to $\boldsymbol{x} = \{\boldsymbol{x}_1, \boldsymbol{x}_2, \ldots, \boldsymbol{x}_T\}$ to produce $\hat{\boldsymbol{h}} = \{\hat{\boldsymbol{h}}_1, \hat{\boldsymbol{h}}_2, \ldots, \hat{\boldsymbol{h}}_T\}$ where the target of $\hat{\boldsymbol{h}}_t$ is $\boldsymbol{x}_{t+1}$ so the difference between $\boldsymbol{x}_{2,\ldots,T}$ and $\hat{\boldsymbol{h}}_{1,\ldots,T-1}$ is used as a loss to control the latent representation of the combiner output $\hat{\boldsymbol{x}}$. Since we are modeling data autoregressively in time, care must be taken with the attention mechanism during training, the transformer uses a modified attention mechanism as described below in Section 3.3.1, Eq. 8.

**Modality Reconstruction.** The autoregressive modality reconstruction models estimate $\prod_{t=1}^{T} p(\boldsymbol{v}_{t+1}, \boldsymbol{a}_{t+1}|\breve{h}_t)$. This is done by applying a separate transformer to $\hat{\boldsymbol{h}}$ to produce reconstructions of the audio and video signals $\hat{\boldsymbol{v}}$ and $\hat{\boldsymbol{a}}$, which is added as an optional loss below. To save on computation, the video input $\boldsymbol{v}$ is down sampled to $\boldsymbol{v}^{small}$ for the reconstruction target, thus the actual reconstruction is $\hat{\boldsymbol{v}}^{small}$.

#### 3.3.1 Attention mechanisms for Autoregressive modeling

Since the autoreggressive models are trained in time, masking is done to satisfy causality. We note that the attention mechanisms within and across chunks need to be modified when masking. This applies to both the Combiner and the Autoregressive learning (Sections 3.2 and 3.3). When masking features for autoregressive modeling, the standard pattern of masking each feature individually would mask features from within the same time-chunk from each other. While this would still satisfy causality, it unnecessarily restricts the model, preventing features from within the same time-chunk from interacting based on position within the time-chunk. To allow features in the same chunk to interact, the autoregressive mask between all features $\boldsymbol{i}$, which fall in a time-chunk $t$, and another feature $\boldsymbol{j}$ is computed as follows ($N$ is the number of features and $T$ the number of time-chunks):

$$mask_j^i = \begin{cases} 0 & j <= \text{ceil}(t * T/N) * N/T \\ 1 & \text{otherwise.} \end{cases} \tag{8}$$

Table 1: Video QA results on MSRVTT-QA. Results in gray show VideoQA as classification.

| Method | Accuracy (%) |
|---|---|
| Just Ask (Yang et al., 2021) | 41.5 |
| ALPRO (Li et al., 2022a) | 42.1 |
| MERLOT (Zellers et al., 2021) | 43.1 |
| VIOLETv2 (Fu et al., 2023) | 44.5 |
| VindLU (Cheng et al., 2022) | 44.6 |
| VideoOFA (Chen et al., 2023b) | 45.4 |
| GIT2 (Wang et al., 2022b) | 45.6 |
| Iterative Co-Tok (Piergiovanni et al., 2022b) | 45.7 |
| VideoCoca (Yan et al., 2022) | 46.3 |
| All-in-one (Wang et al., 2022a) | 46.8 |
| UMT-L (Li et al., 2023) | 47.1 |
| InternVideo (Wang et al., 2022e) | 47.1 |
| Flamingo (Alayrac et al., 2022) | 47.4 |
| M-PLUG2 (Xu et al., 2023) | 48.0 |
| MaMMUT (Kuo et al., 2023) | 49.5 |
| **Ours - TTM** | **50.01** |
| **Ours** | **50.42** |

## 3.4 COMBINING ALIGNED AND NON-ALIGNED AUTOREGRESSIVE MODELING

Text, or other context information, might not necessarily be aligned in time with the video and audio modalities. It is still sequential. So here it is modeled by a separate autoregressive model, devoted to text representations and to combining the visual-audio information together. Assuming tokenization for the input text $t = \{t_1^f, t_2^f, \ldots t_P^f\}$ is provided, obtaining a tokenized text sequence $t = \{w_1, w_2, \ldots w_L\}$ of length L, we model the text sequentially as conditioned on audio and video. In order to combine the outputs of the video/audio autoregressive model we use cross-attention strategy (Alayrac et al., 2022). Here, unlike prior work, all feature representations $\hat{h} = \{\hat{h}_1, \hat{h}_2, \ldots, \hat{h}_T\}$ from the latent causal model are used in the main text model.

$$p(w|\hat{h}) = \prod_{l=1}^{L} p(w_l|w_{l-1}, \hat{h}). \tag{9}$$

The autoregressive text model estimates Eq. 9 by applying a transformer to the input text sequence $w = \{w_1, w_2, \ldots, w_L\}$ and using the latent model output $\hat{h}$ as cross-attention to produce $\hat{w}$. The loss is the standard cross-entropy loss between target $w$ and output text sequences $\hat{w}$. This provides further feedback to the combiner latent representation $h$ through the cross-attention layer. Of note is that since all parts of the model are autoregressive, it is naturally applicable to streaming videos.

## 3.5 MODEL LOSSES

We use the following losses: **Latent space reconstruction loss** is the difference between $x_{2,\ldots,T}$ and $\hat{h}_{1,\ldots,T-1}$ in autoregressive setting such that $\hat{h}_t = x_{t+1}$. We apply a $L^2$ normalization and then take dot product between the feature vectors as the loss (i.e., cosine similarity). **Video reconstruction loss.** Similar to latent space reconstruction above the video reconstruction loss approximates the difference between $\hat{v}^{small}$ and $v^{small}$ also in an autoregressive setting such that $\hat{v}_t^{small} = v_{t+1}^{small}$. We use the same distance metric on the video reconstruction as we use on the latent space reconstruction problem. **Unaligned text cross entropy loss** is the standard cross-entropy loss between $w$ and $\hat{w}$ for the unaligned text output. These losses are weighted to compute the final loss.

## 3.6 IMPLEMENTATION DETAILS

**Model:** Our video+audio model model has 3B parameters; without audio it is 2.9B. A little over half of the parameters are for the audio+video autoregressive model. Our models work on 128 frames customarily, but can handle more for longer videos (e.g., 512). We use 16 chunks and

Table 2: Long video results on ActivityNet and NExT-QA. Gray is for classification setting.

(a) **ActivityNet-QA**.

| Method | Acc % |
|---|---|
| Just Ask (Yang et al., 2021) | 38.9 |
| MERLOT (Zellers et al., 2021) | 41.4 |
| FrozenBiLM (Yang et al., 2022) | 43.2 |
| VideoCoca (Yan et al., 2022) | 56.1 |
| Sing-Temp (Lei et al., 2022) | 44.1 |
| VindLU (Cheng et al., 2022) | 44.7 |
| UMT-L (Li et al., 2023) | 47.9 |
| **Ours - 512 frames TTM** | **49.85** |
| **Ours - 128 frames** | **48.25** |
| **Ours - 512 frames** | **51.13** |

(b) **NExT-QA**.

| Method | (Acc %) |
|---|---|
| CLIP (single frame) | 43.7 |
| VQA-T (Yang et al., 2021) | 52.32 |
| AIO (Wang et al., 2022a) | 50.60 |
| ATP (Buch et al., 2022) | 54.3 |
| VGT Xiao et al. (2022) | 55.02 |
| MIST - CLIP Gao et al. (2023) | 57.18 |
| HiTeA Ye et al. (2022) | 63.1 |
| **Ours - 512 frames TTM** | **73.2** |
| **Ours - 128 frames** | **68.2** |
| **Ours - 512 frames** | **72.0** |

Table 3: **Audio-Video results on Kinetics-Sound, VGG-Sound, and Epic-Sound.**

(a) **Kinetics-Sound**.

| Method | Acc. % |
|---|---|
| MBT (A+V) | 85.0 |
| Ours (Sm, Video) | 81.3 |
| Ours (Sm, A+V) | 85.0 |
| Ours TTM (A+V) | **88.3** |
| Ours (A+V) | **90.1** |

(b) **VGG-Sound**.

| Method | Acc. % |
|---|---|
| UAVM | 65.8 |
| MMT (Zhu et al., 2022) | 66.2 |
| MAViL (Huang et al.) | 67.1 |
| ONE-PEACE | 68.2 |
| Ours TTM (A+V) | 66.4 |
| Ours (A+V) | **69.8** |

(c) **Epic-Sound.**.

| Method | Acc. % |
|---|---|
| SSAST | 53.47 |
| ASF | 53.75 |
| Ours (Audio only) | 62.4 |
| Ours (Video only) | 72.4 |
| Ours TTM (A+V) | **79.4** |
| Ours (A+V) | **78.2** |

Combiner dimension $m = 32$. We apply random masking to the combiner output features at a ratio of $0.75\%$ as a form of dropout regularization as we found this stabilizes the causal model latent reconstruction. Due to the design of our model (partitioning and Combiner), adding more frames, or increasing the chunk size, number of chunks, etc. lead to only marginal increase in parameters. Increasing the number of chunks, while not leading to parameter increases, increases memory, which underscores the importance of the Combiner and particularly the TTM. **Model training:** The model is pretrained on the Video-Text Pairs (VTP) dataset which is collected from noisy video-text pairs from the web (Alayrac et al., 2022). We use only about $12\%$ of the data. All losses are given equal weight during pretraining. During finetuning the unaligned text loss is increased 10-fold to better align the training loss with the final evaluation. The model is trained for 10 epochs on MSRVTT-QA, VGG-Sound, 80 on ActivityNet-QA, 20 on NExT-QA, 8 on Epic-Sound, 4 on Kinetics-Sound.

## 4 EXPERIMENTS

Below we report results on standard Video Question Answering (VideoQA) benchmarks, on long-video VideoQA benchmarks and on Audio+Video benchmarks. We report results using the **open-ended text-generative evaluation**, following (Li et al., 2021b; Lei et al., 2022). Our model generates a free-form text response which is compared to the target response for an exact match. This is more challenging than a classification setting, as our model might generate a correct answer but which is not among the desired classes. This evaluation is more general and widely applicable.

**Video Question Answering.** We first report Video Question Answering results on the MSRVTT-QA VideoQA dataset (Xu et al., 2016), as the most popular Video QA benchmark. The results are shown in Table 1 alongside the best state-of-the-art (SOTA) performances. Our method outperforms the SOTA, including the ones with classification evaluation, and models much larger than ours.

**Long video Question Answering.** We further report Video QA results on long video datasets. ActivityNet-QA (Yu et al., 2019) contains longer videos of about 160 seconds per video. NExT-QA (Xiao et al., 2021) is a targeting complex events with long videos of about 44 seconds. We sample up to 512 frames. Results are in Table 2, showing we outperform the SOTA approaches.

Table 4: **Ablation studies.**

(a) **Effects of proposed components**.

| Model | Frames/Chunks | Acc. |
|---|---|---|
| Baseline | 32/4 | 41.5 |
| + AR | 32/4 | 43.2 |
| + Combiner | 32/4 | 42.1 |
| + AR + Combiner | 32/4 | 44.2 |
| + Pretraining | 32/4 | 45.2 |
| + AR + Comb. + PT | 32/4 | 47.9 |

(b) **Combiner types**.

| Combiner type | Fr./Ch. | Acc. |
|---|---|---|
| Perceiver | 32/4 | 43.1 |
| Transf.+CLS | 32/4 | 43.7 |
| Ours-Transf. | 32/4 | 44.2 |
| Ours-TTM | 32/4 | 44.8 |

(c) **Autoregressive model more frames**.

| Model | Frames/Chunks | Acc. |
|---|---|---|
| Baseline | 64/1 | 41.8 |
| Ours-Autoreg. | 64/8 | 45.1 |
| Ours + BD | 64/8 | 45.1 |
| Ours-Autoreg. | 128/8 | 45.8 |

(d) **Combiner dimension**.

| Model | Fr./Ch. | Dim | Acc. |
|---|---|---|---|
| Ours-8 | 32/4 | 8 | 42.53 |
| Ours-16 | 32/4 | 16 | 43.36 |
| Ours-32 | 32/4 | 32 | 44.20 |
| Ours-64 | 32/4 | 64 | 44.22 |

**Audio-Video Results.** Table 3 shows results on three Audio-Video benchmarks: Kinetics-Sound (Arandjelovic & Zisserman, 2017), VGG-Sound (Chen et al., 2020) and Epic-Sound (Huh et al., 2023). Since these datasets are Audio-Video classification, we treat the task as open-ended generation: we input the text 'Classify the video audio clip.' and expect the output to be the target class name e.g., 'playing drums'. Across all datasets, we outperform the SOTA with large margins, despite the more challenging open-text generation evaluation. Methods are MBT (Nagrani et al., 2021) One-Peace (Wang et al., 2023), SSAST, ASF (Huh et al., 2023), UAVM (Gong et al., 2022).

### 4.1 ABLATIONS

The ablations (Table 4), are conducted with the video and text model in order to understand the main behaviors of this architecture. We also use a smaller model and configuration, and where applicable, 2x fewer pretraining steps with the same batch size to save compute (details are in the Appendix).

**Main model components:** We start with ablations where we study the effect of each component (Table 4a). We find that on top of a baseline model, adding each part, the autoregressive (AR) model, the combiner, and pretraining, each individually help and the combination of all three further help.

**Combiner type ablations:** We compare the Combiners: transformer-based (ours, CLS and Perceiver (Alayrac et al., 2022)) and TTM. The CLS-token inspired combiner appends $m$ learnable features to the end of the sequence and takes their values as the combined features after passing the whole sequence through the transformer. These are visualized in Figure 3. We use the same settings for direct comparison. Table 4b shows that our proposed combiners perform best.

**Autoregressive modeling in time:** We ablate the Autoregressive part of the model. Table 4c shows that processing the video in chunks autoregressively in time is more advantageous than learning from the full video at once, with a large jump in performance (first two rows). Not only is our autoregressive model feasible for longer videos but it is also more beneficial for the same size inputs. More frames per chunk contribute to the improvements (rows two and four). We also compare to a bidirectional (BD) model, finding that the performance is the same as the autoregressive portion.

**Combiner size ablations.** We further compare the number of features output by the combiner per timestep. We noticed a trend for larger combiner outputs giving better results, lines 3-4 (Table 4d). We chose 32 as a trade-off between sufficiently compact feature length and sufficiently expressive.

## 5 CONCLUSIONS

We propose a multimodal autoregressive model which decouples the autoregressive modeling into a component, devoted to time-aligned modalities (video, audio) and another one for the non-aligned, contextual modalities (text). To address long video/audio inputs we propose to partition the media inputs and learn from them jointly by a Combiner, which allows to control the sequence lengths. The model can handle 512 frames, without increasing its size. Our approach not only enables working with long videos effectively but also outperforms SOTA, achieving gains over much larger models.

## 6    ETHICS STATEMENT

The model described is trained on video/audio and text data which might be noisy and with inaccurate labeling. This might propagate inaccuracies or biases into the model. We have used the model for evaluation purposes and to compare to the established benchmarks in the field. We evaluate on publicly available datasets, which have been previously used for evaluation and benchmarking purposes. No new datasets are proposed or studies with use of human subjects are conducted.

## 7    REPRODUCIBILITY STATEMENT

The proposed model is based on transformer model architectures, which are widely used and available. The model implementation details and pretraining specifics are described in Section 3.6. The model architecture configurations are further described in Section A.3. We note that other model configurations are also possible and, especially, constructing larger models can produce even better results; we also experimented with a smaller model for comparison in Table 3a and in the ablation studies, Table 4; its details are in Section A.3. We have described the parameters for subdividing the media inputs in Section 3.6. The parameters of the Combiner components, e.g., dimensions, outputs are also described in Section 3.6.

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

# A  APPENDIX

## A.1  DATASETS DETAILS

The following datasets have been used for evaluation in the paper:

MSRVTT-QA (Xu et al., 2016) is a popular Video QA dataset of about 10K video clips and 243K question-answer pairs. It is derived from the MSRVTT dataet by automatic question-answer pairs and contains a certain level of noise. Videos are about 14 seconds in length, on average.

ActivityNet-QA (Yu et al., 2019) is a commonly used benchmark for understanding of longer videos. It contains 5,800 videos and 58,000 question-answer pairs. It has much longer videos which entail longer and more complex scenes. The video length is about 160 seconds per video on average.

NExT-QA (Xiao et al., 2021) dataset is also addressing long video understanding. It contains 5,440 videos and about 52K manually annotated question-answer pairs. The average length of the videos is 44 seconds. Apart from questions related to descriptions and in the video, NExT-QA focuses on questions related to events and sequence of events within the video, e.g., causal ('Why' and 'How' questions), and temporal - questions related to order of events, or related to concurrent activities and others.

VGG-Sound (Chen et al., 2020) is a large-scale audio-video dataset, featuring over 200,000 videos accompanied by audio sounds. The data is formulated as classification tasks with 300 audio classes.

Epic-Sound (Huh et al., 2023) is an audio-video dataset based on the Epic-Kitchens dataset. It has 78.4k examples and 44 target classes.

Kinetics-Sound (Arandjelovic & Zisserman, 2017) is a dataset derived from the popular Kinetics-400 video recognition dataset. Kinetics-Sound includes audio inputs sampled together with the video and has 36 classes.

All the abovementioned audio-video datasets used in the paper, have been formulated as datasets for classification tasks. Here we use the class outputs (which are typically short phrases describing an activity, instrument or type of sound e.g 'Knocking on a door') and treat them as open-ended text generation tasks and thus they are now audio-video-text datasets.

## A.2 COMBINER VISUALIZATIONS.

In Figure 3, we visualize the different combiners we explored. The Transformer combiner, CLS combiner and Perceiver combiner are all based on transformers taking input of all the video + audio features and reducing them to $m$ combined features. We found our main combiner to outperform the other two in Table 4b. We note that the Perceiver combiner is an adaptation of our combiner by applying Perceiver resampling (Jaegle et al., 2021). The TTM combiner is conceptually different: rather than taking all the previous features as input, it takes only the current timestep features as input and uses a memory mechanism with read and write operations to update it. It then uses a MLP to produce the $m$ combined output features. This reduces memory and compute use and sometimes reduces accuracy.

## A.3 ADDITIONAL MODEL AND IMPLEMENTATION DETAILS

**Model Details.** The autoregressive text model contains about 1.3B parameters, 400M are for cross-attention weights and 400M for the vocab embeddings and following specifications: layers=18, model dims=1536, hidden dims=12288, heads=12, and head dims=128. About 100M parameters are for the additional weights associated with audio. The remaining parameters are for the video input processor, combiner, causal latent model and video reconstruction model (a bit over 1.5B parameters in total). The combiner, causal latent model and video reconstruction model are transformers with 128M parameters and the following specifications: layers=8, model dims=1024, hidden dims=4096, heads=16, and head dims=64. The video chunk processor has roughly 630M parameters, following ViT-Huge. The convolutional tubes have 1.5M parameters and the transformer has 630M parameters and following specifications: layers=32, model dims=1280, hidden dims=5120, heads=16, and head dims=80. The total parameter size is 3B parameters.

The smaller model used for ablations keeps the same combiner, causal latent model, and video reconstruction model as the main model. However the autoregressive text model is reduced to 128M parameters with the same settings as the combiner, and has 20M cross-attention weights and 260M parameters for the vocab embedding. The audio parameters are held roughly the same. The video input processor is reduced to ViT-Large which has 300M parameters and the following specifications: layers=24, model dims=1024, hidden dims=4096, heads=16, and head dims=80. The total parameter size is 1.15B parameters.

The TTM Combiner, as mentioned implemented by a TokenLearner (Ryoo et al., 2021) function and a transformer. The output dimension $K = 32$ is the same as the output dimension for the standard Transformer Combiner. The output dimensions for the 'Read' and 'Write' functions are 512 and 256, respectfully. These two parameters can be controlled independently to allow more or less capacity to the TTM Combiner. The transformer used within the 'Process' function is of 2 layers, 128 hidden dimension and 12 heads. These are fixed throughout the paper.

**Model Pretraining.** The pretraining data is the Video-Text Pairs (VTP) dataset which is collected from noisy video-text pairs from the web (Alayrac et al., 2022). The main pretraining is done for the autoregressive, combiner, and the learning components processing the low-level video features (e.g., video tubes convolutions). The model's image and text backbones and cross attention layers are initialized from a contrastively image-text pretrained model, as in CLIP. The text backbone is frozen during pretraining while the other components including the cross attention weights are unfrozen. During pretraining, the combiner model, causal latent reconstruction model and video reconstruction model and video tubes are all randomly initialized. All losses are given equal weight

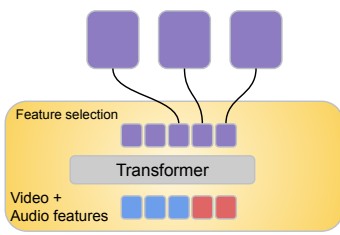

(a) **Transformer combiner.** All features are input to the transformer, a smaller number of *m* features are selected as combined features.

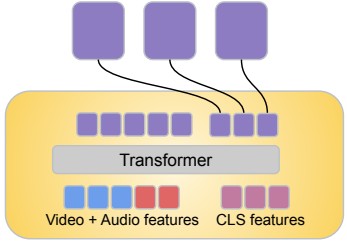

(c) **CLS combiner.** Adds *m* features to the input, then after the transformer, takes those *m* features as the combined features.

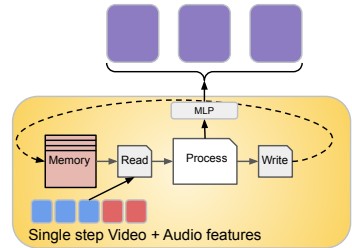

(b) **TTM Combiner.** Uses the TTM mechanism to store a memory and compute the *m* combined features for each time step. This process is repeated for each time step.

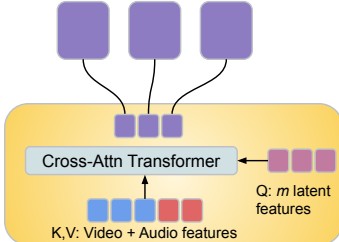

(d) **Perceiver combiner.** Adds *m* latent features for cross attention to the transformer, resulting in *m* outputs.

Figure 3: Visualization of the different combiners we explored in this paper. The Transformer combiner, which is the main one we used, simply takes the last $m$ features of the output to represent the combined inputs. We found this to work well. The CLS combiner and Perceiver combiner we found both underperformed the base combiner. The TTM combiner is different, it uses a memory to store the previous representations and has read, process and write operations. We found this method saved memory with some tradeoff for accuracy for some datasets.

during pretraining. For pretraining, we used a learning rate of $1 \times 10^{-5}$, batch size of 32, image resolution of $224 \times 224$, 128 frames.

**Fine-tuning.** During finetuning all parameters are unfrozen. In addition the unaligned text loss is given extra weight and increased 10-fold to better align the training loss with the final evaluation, since the latent space and video reconstruction are not evaluated. The model is trained for 10 epochs for the MSRVTT-QA dataset and for 80 epochs on ActivityNet-QA and 20 epochs on NExT-QA. For these datasets, we finetune with a learning rate of $5 \times 10^{-6}$, weight decay of $0.01$, image resolution of $448 \times 448$, batch size of 32. We use 128 frames for the main experiments, except for the long video benchmarks where we also report performance with 512. Sampling more frames from the other benchmarks is not porductive as they contain relatively short videos. We used dropout of 0.1, label smoothing of 0.2

**Video-Audio Implementation Details.** Since the model is pretrained on VTP data which most videos lack audio, we try adding a further audio pretraining step here. We use AudioSet-2M (Gemmeke et al., 2017) and train the model to output the text of the class names. In this step, we freeze the weights of the model, other than the audio weights, allowing the model to learn to handle the spectrogram inputs. During fine-tuning on the eval datasets, we fully train the model. During finetuning, we also use Mixup (Zhang et al., 2017), specaugment (Park et al., 2019), dropout and label smoothing, following the settings of previous works (e.g., (Georgescu et al., 2022)). We use a learning rate of $1 \times 10^{-5}$, with the Adam optimizer (default settings), weight decay of $0.0001$, cosine learning rate decay. We use an image resolution of $448 \times 448$, batch size of 32, and 128 frames.

**Ablation experiments details.** The ablation experiments in Tables 4a, 4b, 4c, 4d are conducted with our small model. The Baseline in Table 4a uses partitioning, as the rest of the approaches tested in the table, and concatenation of the features to be maximally comparable to others. Experiments without partitioning are in Tables 4b, 4c, 4d.

