# OpenReview forum: "Partition and Conquer: A Multimodal Autoregressive Model for Time-Aligned and Contextual Modalities"
_ICLR.cc/2024/Conference — ICLR 2024 Conference Withdrawn Submission_

### Official Review · Reviewer_ypJZ · 2023-10-31

**Soundness:** 2 fair
**Presentation:** 2 fair
**Contribution:** 2 fair
**Rating:** 3
**Confidence:** 3

**Summary:**

This paper addresses the challenge of multimodal learning and proposes a solution by dividing the modeling process into separate autoregressive models for different modalities. They introduce a multimodal model that handles time-synchronized and sequential modalities, partitioning long sequences into snippets. A Combiner mechanism extracts and fuses audio-video features, generating compact representations per snippet. The approach achieves state-of-the-art performance on multimodal benchmarks, addressing computational demands by learning compact representations and capturing temporal dependencies.

**Strengths:**

1. The proposed approach achieves state-of-the-art performance in experimental results, demonstrating its effectiveness.
2. The paper highlights the differences between time-aligned and non-time-aligned data, providing valuable insights into the challenges of multimodal learning.

**Weaknesses:**

1. The paper does not fundamentally address the issue of long sequences but rather provides a layer of compression. However, this compression technique is already commonly used in existing representation learning methods.
2. The description and illustration of the Combiner mechanism are inconsistent. The Combiner in the figure only takes the audio and video features of the current chunk as input, which contradicts the textual description and Equation 2. This inconsistency can be confusing for readers.
3. The method lacks innovation, as the design of the attention mechanism is not novel and resembles the commonly used chunkwise attention [1].
4. The model’s loss function design is unclear. For example, it includes video reconstruction loss but lacks audio reconstruction loss, which raises questions about the rationale behind this choice.

[1] Chiu C C, Raffel C. Monotonic Chunkwise Attention[C]//International Conference on Learning Representations. 2018.

**Questions:**

The same as weaknesses.

---

### Official Review · Reviewer_KVWc · 2023-11-01

**Soundness:** 2 fair
**Presentation:** 2 fair
**Contribution:** 2 fair
**Rating:** 3
**Confidence:** 3

**Summary:**

This paper presents a novel multi-modal pertaining scheme. A module is dedicated to auto-regressively predict the future video and audio chunks from past chunks. A combiner module is proposed to get a compressed joint representation of the audio-video features. Then the original audio/video features are predicted from the output of combiner in an auto-regressive manner. Additionally, the latent features from audio-video encoder are used to influence the auto-regressive prediction of text label/captions by using a cross-attention mechanism. The model is trained end-to-end using reconstruction losses for audio/video features and cross entropy loss for text output. They also modify the attention mechanism to accommodate features from future time-steps belonging to same chunk.

**Strengths:**

1. Unlike previous works in multi-modal learning with video and text that use less features, authors propose a model that works with video features extracted with better time resolution (128 or 512 frames vs 8/32 frames).
2. They have shown the effects of using various types of combiners which is one of their propose module. Along with that they also looked at the effects of chunking in their ablation study. Hence, they showcased the effectiveness of these two proposed modules.
3. Authors have considered various datasets for multi-modal tasks such as video QA, long video QA and audio classification to show the performance of their proposed framework.

**Weaknesses:**

1. In the proposed framework, authors first compute a latent representation “h” from combiner representation “x” before reconstructing the actual audio/video features “(a,v)”. It is not well-motivated why the combiner representation can’t act as latent representation and be used to reconstruct a,v directly from it. What is the added benefit of introducing another latent representation h?
2. The proposed model has about 4B parameters. It would have been interesting to show the knowledge it learned during pertaining by performing zero-shot/few-shot experiments rather than/along with full-scale fine-tuning.
3. The model comparisons do not seem fair in experiments section. The proposed model has about 4.3B parameters and authors do a fine-tuning on the downstream datasets after pre-training. In comparison, one of the baseline models UMT-L has only about 300M parameters. Moreover, it was fine-tuned for only 10 epochs on ActivityNet while the propose model is fine-tuned for 80 epochs on the same dataset. Similarly it seems FrozenBiLM for ActivityNet and Flamingo for MSRVTT were implemented in a zero-shot fashion without fine-tuning in their original papers. MAMMUTT used for MSRVTT has image encoder of size 650M params while in this case the video encoder is of size ~3B parameters. Similarly models MMT and MAViL used for audio tasks have also parameters in the range of ~100M. Authors should mention the model parameters and whether they were fine-tuned along with accuracies for fair comparison.

**Questions:**

1. Effect of vanilla attention vs modifying their attention to accommodate features from future testers belonging to same chunk
2. What is the tubeless dimension used to extract video features? What are the spectrogram dimensions?
3. Authors mention using 128 frames or 512 frames for their experiments. How did they sample these frames from the video say which has N frames sampled at some pre-defined fps where N > 128 or 512?
4. Was the random masking applied only to combiner features while predicting latent features? Did the authors try applying random masking to original audio/video features to get the combiner feature or apply masking to hidden features while reconstructing the original audio/video features? Masked auto encoders have been shown to benefit AV tasks and hence applying masking at various stages could lead to the model learning effective representations.
5. I am not sure if I understand how the chunking and combiner works as mentioned in section 3.6. It seems the number of input frames is 128 which is divided into 16 chunks. So each chunk has 128/16 = 8 frames for each modality. So in total the input to combiner for each chunk is 16 frames (8 from video concatenated with 8 from audio). The combiner output dimension is set at m=32. One of the motivations to use combiner was to compress the representation betaking last few frames of the join representation but in this case 32 > 16.
6. In equation 8, shouldn’t “t” refer to the actual time-step index rather than the time-chunk index?

---

### Official Review · Reviewer_AMDB · 2023-11-06

**Soundness:** 3 good
**Presentation:** 3 good
**Contribution:** 2 fair
**Rating:** 5
**Confidence:** 3

**Summary:**

This paper proposes a model that separately models the different modalities such as video, audio, and text via their separate focused autoregressive models. A combiner model fuses the information learned from audio-video with text. To handle long videos, the paper proposes to chunk audio-video signals to extract high-level information and reduce the number of tokens. Extensive experiments are done to study the impact of various hyperparamers. The proposed model achieves consistent and impressive improvements on various benchmarks.

**Strengths:**

1) The paper is well-written and explains the proposed ideas in detail.

2) The proposed model uses components that are best suited to model various modalities to maximize the information extracted from the inputs.

3) The model shows good performance improvements over state-of-the-art models. Evaluation across a wide variety of datasets shows the effectiveness of the proposed model.

**Weaknesses:**

1) None of the tables compare the parameters of the proposed model with previous models. The paper claims multiple times that their model outperforms much larger models. Including the parameters of the previous models will help back this claim and also show that the performance improvements are due to the proposed modifications and not simply a large model. For example, the UMT-L model [r1] (Table 2a) contains 304 million parameters and the HiTeA model [r2] (Table 2b) contains 17 million parameters. Both of these models contain a significantly lower number of parameters than the proposed model, 3 billion parameters.

2) On a similar note, a small table showing the number of video-text pairs dataset sizes (e.g. Table 6, 11 in [r1]) used for training the proposed model and the previous models would help highlight the importance of the proposed model. It is unclear whether the improvements are due to scaling i.e. larger models, bigger datasets, or due to the proposed modifiaction such as Comiber, using separate models for time-aligned and unaligned modalities.

3) More details should be included about the ablation studies. For example, what datasets are used for training and evaluating the model used in ablation studies? How does the smaller model compare with previously proposed models with a similar number of parameters?


r1) Kunchang Li, Yali Wang, Yizhuo Li, Yi Wang, Yinan He, Limin Wang, and Yu Qiao. Unmasked teacher: Towards training-efficient video foundation models. In ICCV, 2023.
r2) Qinghao Ye, Guohai Xu, Ming Yan, Haiyang Xu, Qi Qian, Ji Zhang, and Fei Huang. Hitea: Hierarchical temporal-aware video-language pre-training. arXiv preprint arXiv:2212.14546, 2022

**Questions:**

1) How long does training the model take? What computational resources are used for training the model?
2) Would using a pretrained model for audio improve the downstream performance?